# Transesophageal Echocardiography-Guided Transseptal Left Atrial Access to Improve Safety in Patients Undergoing Pulmonary Vein Isolation

**DOI:** 10.3390/jcm11092546

**Published:** 2022-05-01

**Authors:** Rahel Zuercher, Anique Herling, Marc T. Schmidt, Marta Bachmann, Stephan Winnik, Firat Duru, Urs Eriksson

**Affiliations:** 1Division of Cardiology and Electrophysiology, GZO Regional Health Center, 8620 Wetzikon, Switzerland; rahel1995@hotmail.com (R.Z.); anique.herling@usz.ch (A.H.); email@marcschmidt.net (M.T.S.); marta.bachmann@usz.ch (M.B.); stephanh.winnik@gzo.ch (S.W.); 2Division of Electrophysiology, Department of Cardiology, University Heart Center, 8091 Zurich, Switzerland; firat.duru@usz.ch; 3Faculty of Medicine, University of Zurich, 8006 Zurich, Switzerland

**Keywords:** atrial fibrillation, pulmonary vein isolation, transseptal puncture, transesophageal echocardiography, patient safety

## Abstract

Background: Endovascular pulmonary vein isolation (PVI) has become an important strategy for rhythm control in patients with symptomatic atrial fibrillation (AF). Transseptal access is a critical step of this procedure and can result in potentially life-threatening complications. This retrospective study evaluates the safety of standardized, transesophageal echocardiography (TEE)-guided transseptal access to the left atrium in consecutive patients who underwent PVI. Methods: After the implementation of a standardized, TEE-guided procedure for transseptal access, the data of 404 consecutive PVI procedures using radiofrequency ablation and 3D-mapping were prospectively collected over 5 years. TEE-guided transseptal punctures were performed on 375 patients undergoing one to three PVIs. The patient cohort was retrospectively analyzed for major and minor complications, fluoroscopy time, fluoroscopy dose and ablation outcomes. Results: No single complication related to transseptal access occurred, affirming the safety of the TEE-guided approach. Fluoroscopy time and fluoroscopy dose decreased significantly after 152 procedures. PVI-related minor complications occurred in 11 procedures (2.6%) and included 10 vascular-access-related complications (2.4%) and 1 TEE-related esophageal hematoma (0.2%), which healed spontaneously. Conclusion: Our single-center study shows that TEE guidance may allow safe transseptal access to the left atrium in patients undergoing PVI.

## 1. Introduction

Atrial fibrillation (AF) is the most common cardiac arrhythmia with a lifetime risk of 22–26% and an overall prevalence of 3% [1]. The prevalence of AF is particularly high in elderly people or in patients with comorbidities such as coronary heart disease, hypertension, heart failure, valvular heart disease, diabetes mellitus, obesity or chronic kidney disease. AF is associated with an increase in all-cause mortality, hospitalizations and higher morbidity including stroke, thromboembolism and heart failure, as well as an impaired quality of life [2]. Therapeutic options in AF range from medical therapy for rhythm or rate control to interventional approaches including pulmonary vein isolation (PVI). In patients with paroxysmal AF, PVI has been shown to be more effective than antiarrhythmic drug therapy with similar overall complication rates [2,3,4]. Recently, the CASTLE-AF study showed that PVI is superior to medical therapy in patients suffering from heart failure [5].

PVI is generally regarded as a safe procedure, but it carries a low but relevant risk for potentially life-threatening complications. In the near future, a further reduction in complication rates could be achieved by implementing standard operating procedures (SOPs) in high-volume centers. Nevertheless, PVI includes several critical steps, in particular, transseptal access to the left atrium [6,7,8], which can be associated with serious complications, such as cardiac tamponade, aortic root puncture or systemic embolism. According to the current literature, the total complication rate of PVI varies between 4.5% and 6%, with major complications [4] such as cardiac tamponade, stroke or transient ischemic attacks, pulmonary vein stenosis, or esophageal injury leading to atrio-esophageal fistula within weeks after ablation [2].

In this study, we aimed to investigate the safety of transesophageal echocardiography (TEE)-guided transseptal puncture, which is integrated as an SOP during PVI procedures in the setting of a medium volume electrophysiology laboratory affiliated with the Zurich University Hospital.

## 2. Methods

### 2.1. Study Design

This is a retrospective study, which included symptomatic patients with paroxysmal, persistent or long-standing persistent AF who underwent TEE-guided transseptal access during PVI (or Redo PVI) procedures at the GZO—Regional Health Center in Wetzikon, Switzerland.

The only exclusion criterium was refusal to provide written consent to data collection. All procedures were performed under general anesthesia according to a strict SOP protocol. All included patients gave written informed consent for general anesthesia, TEE and PVI and agreed to the anonymous evaluation of their data for research and quality control. The study protocol and data evaluation was retrospectively approved by the Cantonal Ethics Committee (KEK project ID 2018-00451, Zurich, Switzerland).

Patient data were extracted from the clinic information system and collected in a secure web-based database (RedCap). In accordance with the Ethics Committee decision, all data are available for research purposes as encrypted anonymous files upon request. 

### 2.2. Anesthesia

PVI procedures were performed under general anesthesia on intubated and ventilated patients (volume-controlled ventilation mode for optimal 3D-electromagnetic mapping). Anesthesia was induced and maintained with propofol and remifentanil. For anesthesia induction and TEE probe placement, rocuronium (1 mg/kg) was administered. The mean arterial blood pressure was kept at 70 mmHg.

### 2.3. 2D-TEE Procedure

2D-TEE was performed under general anesthesia on intubated and ventilated patients with either an Acuson SC2000 (Siemens) or an EPIQ 7 (Philips) ultrasound system. Exclusion criteria for TEE were esophageal disease with known stricture, tumor, varices or prior esophageal and stomach surgery < 8 weeks ago. Barrett esophagus, diverticula or aneurysms of the descending aorta were not considered contraindications.

The TEE probe was blindly inserted through a mouth guard and gently advanced to the esophagus under the guidance of the index and long finger of the operator’s hand. Laryngoscopy was used if two attempts of blind insertions failed.

Prior to the ablation procedure, a complete TEE examination, including duplex sonography and a “microbubble” test, was performed on all patients to assess cardiac function, valves and aorta, and to exclude atrial thrombi and to check for a patent foramen ovale. Each TEE was performed by one of the electrophysiologists familiar with echocardiography, left atrial anatomy and ablation workflow. 

Catheter access to the left atrium was guided by 2D-TEE in all procedures. If a patent foramen ovale was detected, an SL0 transseptal sheath was inserted through the foramen ovale; otherwise, transseptal puncture was performed under TEE guidance, as illustrated in Figure 1.

For TEE-guided transseptal access, the operator slowly retracted the SL0 transseptal sheath aligned along the interatrial septum in a 90–110° bicaval TEE view until it dropped onto the septum and tenting in the direction of mid left atrium became evident (the optimal view was usually 40–70°). Next, the needle was advanced and directed through the septum, keeping the maximal possible distance to the aortic root as well as to the posterior wall of the left atrium. Importantly, the TEE operator continuously kept the sheath and needle visible for the operator during the entire procedure (Figure 1). Interaction between TEE and the EP operator is essential. 

Heparin was administered at a dose of 50 IU per kg body weight immediately after venous access, and another 50 IU per kg was given after transseptal passage of the introducer sheath to the left atrium. Heparin was further administered to keep the activated clotting time (ACT) at a target range of 300–350 s. 

The TEE probe was removed after the exclusion of procedure-related pericardial effusion prior to radiofrequency ablation (RF).

### 2.4. Ablation Procedure

A CT (Computed Tomography) image of the left atrium was imported into a 3D-electroanatomic mapping (EAM) system (CARTO, Biosense Webster, Diamond Bar, CA, USA) using custom-designed software (Carto-Merge, Biosense Webster, Diamond Bar, CA, USA). Radiofrequency ablations were performed with an irrigated steerable and pressure-sensitive ablation catheter (Thermocool Smartouch D/F curve, Biosense Webster, Diamond Bar, CA, USA) with a power output between 25 and 30 Watts, as previously reported [4]. Successful isolation was ensured by the demonstration of both the entrance and exit being blocked from the PVs. 

### 2.5. Postinterventional Monitoring

Patients were monitored with telemetry for 24 h and were discharged the next day. After the procedure, all patients received oral anticoagulation. Patients were given proton pump inhibitors for four weeks daily to prevent atrio-esophageal fistulas. All patients were followed up at three months after the procedure, including 48 h ECG recordings. In the case of early recurrences, electrical cardioversion was performed to support reverse atrial remodeling.

### 2.6. Data Collection

Patient characteristics (age, sex, body-mass index (BMI), type of AF (paroxysmal, persistent or long persistent), EHRA scores, anticoagulation prior to intervention and CHA_2_DS_2_-VASc scores as listed in Table 1 were collected. Complications were prespecified as summarized in Table 2 and reported as venous-access-related, transseptal-access-related, ablation-related and TEE-related (Table 2). Major complications were death, myocardial infarction, stroke or emergency vascular, cardiac or visceral surgery for any cause during a follow-up period of 3 months after the PVI procedure. In addition, postintervention data on AF recurrence rates were recorded for all patients.

### 2.7. Statistics

Data were analyzed in a Jupyter notebook using the pandas library. Data were compiled and displayed in Excel. Diagrams were created with Excel. The results are presented using descriptive statistics. For categorical variables, the number and percentage of participants are presented. For continuous variables, the results include number, mean ± SD and 95% bilateral confidence intervals, where pertinent. Continuous variables were compared using the paired Student’s *t*-test. A *p* value of <0.05 was considered significant. All analyses were performed using SPSS 24.0 software (SPSS Inc., Chicago, IL, USA).

## 3. Results

The data from 404 consecutive PVI interventions with 375 transseptal punctures were studied. The characteristics of the study population are shown in Table 1. The average age of the patient population was 65.8 ± 0.6 years, and 110 (38.9%) of these were females. Paroxysmal AF was present in 79.2% of the patients. A total of 302 (79.2%) of the PVIs were primary interventions, 102 (20.8%) interventions were re-do procedures. CHA_2_DS_2_-VASc scores, as listed in Table 1, reflect the co-morbidities of the study population. Most patients (81.4%) were anticoagulated with factor X antagonists prior to the PVI. 

There were no major anesthesia-related complications in the patient cohort, such as anaphylaxis, respiratory depression/hypoxic brain injury, endotracheal tube displacement or re-intubation, aspiration, malignant hyperthermia, nerve injury or cardiac arrest.

Twelve complications (2.8%), as prespecified in the Methods section, were counted in the study population (Table 2), with none of them being life-threatening over a follow-up >3 months. Moreover, there were no transseptal-access- or ablation-related complications. Most minor complications were related to venous access, including arteriovenous fistula, femoral pseudoaneurysms and one retroperitoneal hematoma. Moreover, thrombosis of the femoral vein was attributed to the PVI in one patient, and one patient developed a small esophageal hematoma, which was considered as a prespecified TEE-related complication. The esophageal hematoma was diagnosed via CT scan two days after the procedure in a patient complaining dysphagia. It healed spontaneously without sequelae during follow-up. There was also no single case of damage to teeth, which could be attributed to the TEE procedure as well as to anesthesia. 

Minor complaints, such as a sore throat and small lacerations to lips, tongue, gums or throat were not specifically addressed in this study because they can result from both anesthesia and the insertion of a TEE probe.

For all 404 procedures, the mean fluoroscopy time was 5.825 ± 5.07 min (mean ± SD) and the mean fluoroscopy dose was 642.4 ± 612.8 cGyxcm^2^ (mean ± SD). Importantly, mean fluoroscopy times and fluoroscopy doses markedly decreased after 152 interventions, reflecting the interactive learning curve of the operators (Figure 2 and Figure 3).

## 4. Discussion

Our findings demonstrate that standardized, TEE-guided transseptal access in PVI procedures was safe, and the overall complication rate in our cohort was quite low and mainly due to venous access. The current literature reveals broad diversity in the study design, definition, and analysis of complication rates for PVI. Thus, a direct comparison between different studies is difficult. Even major and minor complications are neither uniformly defined nor classified. Our study describes the safety of a well-defined, uniform PVI procedure in the setting of an intermediate volume electrophysiology laboratory. 

The overall complication rate of 2.8% in our cohort is low as compared to other previously published reports [7,9,10,11,12,13]. In fact, studies reporting low complication rates, as published by De Ponti et al. [7] or Matoshvili et al. [10], for example, did not address overall PVI-related periprocedural complications, but included transseptal-access-related complications alone. In fact, transseptal-puncture-related complications represent the most prevalent major PVI-associated risk in all studies. These complications, however, were completely absent in our cohort. We therefore believe that TEE-guided transseptal access in general anesthesia may become an appreciated standard operation procedure to improve patient safety in PVI procedures.

Cerebrovascular events such as stroke or transitory ischemic attack are among the major complications of PVI and are associated with relevant morbidity and mortality [14]. In the literature, these complication rates range from 0.5% to 1% [10,11,12,14]. No major complication, and particularly no symptomatic cerebrovascular event, occurred in our study. Nevertheless, it must be admitted that we do not perform routine cerebral magnetic resonance imaging (MRI) on asymptomatic patients after PVI. During one intervention, a thrombus was identified via TEE on the mapping catheter. The intervention was interrupted, and additional heparin was administered. Thrombus dissolution was monitored via TEE, and the intervention was safely continued without any neurological deficits and without lesions in two brain MRI scans immediately after and 24 h after PVI. This critical incident impressively demonstrates the additional safety obtained using TEE guidance. 

It is important to note that TEE itself is also an invasive diagnostic procedure and carries a certain—albeit small—risk [15], especially during cardiac surgery [16] or interventional cardiology [17]. In our cohort, we observed one TEE-related complication, an esophageal hematoma, which healed without sequelae and needed no additional interventions. 

In our cohort, the majority of the complications (10 out of 12) resulted from venous access. The implementation of ultrasound-guided vascular access could reduce such complications as well, and therefore, we have currently integrated ultrasound-guided puncture in our SOPs [18,19,20].

Our study population is representative and comparable to the PVI populations of other trials, including patients with symptomatic, mostly paroxysmal AF, as well as low to moderate reduction in daily performance capacity, who carry a low perioperative risk during general anesthesia [21]. However, as CASTLE-AF [5] has recently demonstrated, PVI gains increasing importance for AF management for patients with heart failure and a consecutively higher perioperative risk. In order to offer a minimal risk procedure to such higher risk patients, procedure-related SOPs need to be improved and adjusted continuously. 

As TEE-guided transseptal access is standard procedure in our hospital, this study did not include a control group for comparison. Consequently, our findings can only be discussed in the context of data from published trials. Moreover, standardized, TEE-guided transseptal access requires cooperation between the TEE operator and the electrophysiologist performing the transseptal access. Whether our findings can be translated into a setting where dedicated imaging specialists not familiar with electrophysiology perform TEE remains to be answered.

As illustrated in Figure 2 and Figure 3, the implementation of TEE-guided transseptal access successively and markedly reduced fluoroscopy times and doses. Nevertheless, a learning curve reflecting individual and team proficiency was evident. Meanwhile, our operators successively reduced fluoroscopy to minimal radiation exposure. In fact, during the last year of the study period, it even became possible—in selected cases—to place coronary sinus and ablation catheters with the combined help of the electromagnetic mapping system and TEE without any fluoroscopy.

Taken together, our single-center study shows that TEE guidance may allow safe transseptal access to the left atrium in patients undergoing PVI. 

## Figures and Tables

**Figure 1 jcm-11-02546-f001:**
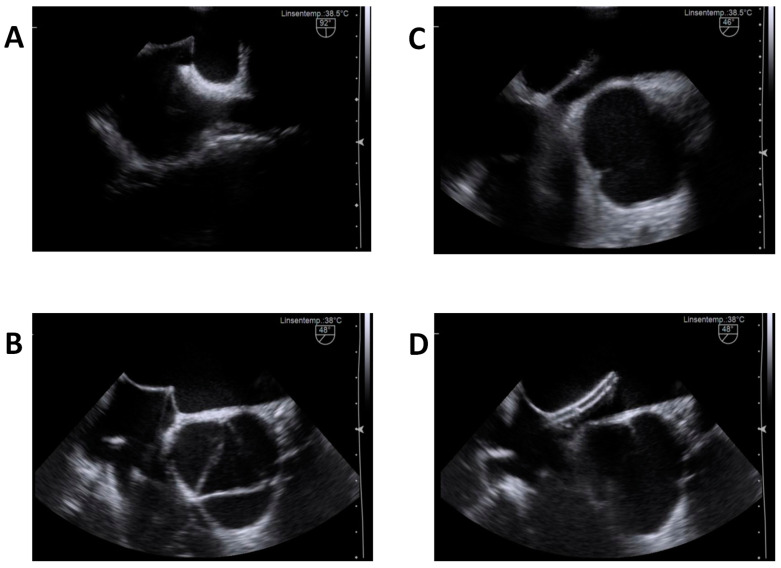
Illustrative TEE—views of the transseptal puncture procedure. (**A**) shows septal tenting immediately after retraction of the sheath in the bicaval view (90–110°, usually). (**B**) shows the advanced needle aligned in an optimal direction, away from the aortic root, as well as within acceptable distance to the posterior left atrial wall. (**C**) catches the instance of septum perforation in the same patient. (**D**) illustrates the advanced introducer sheath within the left atrium.

**Figure 2 jcm-11-02546-f002:**
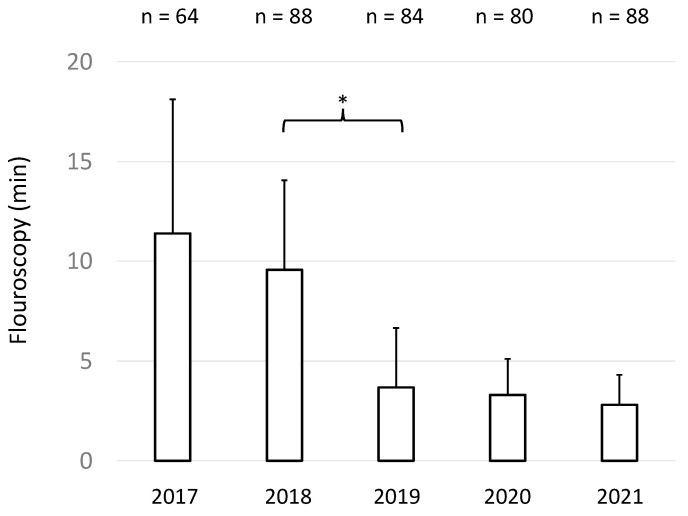
Mean fluoroscopy time (minutes) ± SD of the procedures of each year over a period of five consecutive years. * indicates *p* < 0.05.

**Figure 3 jcm-11-02546-f003:**
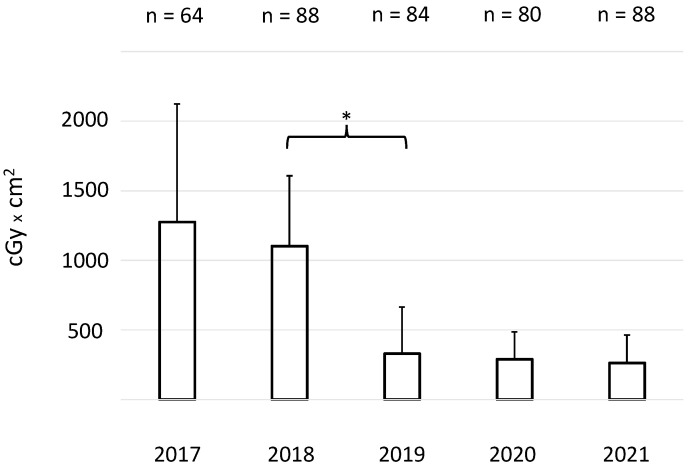
Mean X-ray doses (cGy × cm^2^) ± SD of the procedures of each year over a period of five consecutive years. * indicates *p* < 0.05.

**Table 1 jcm-11-02546-t001:** Characteristics of 404 patients undergoing transseptal access for pulmonary vein isolation.

Demographics			
Age (years, mean ± STD)		65.8 ± 0.6	
Gender (% female)		38.9%	
BMI (mean ± STD)		28.2 ± 0.3	
Atrial fibrillation			
Paroxysmal	320		(79.2%)
Persistent	71		(17.6%)
Long persistent	13		(3.2%)
CHA_2_DS_2_-VASc			
0	59		(14.6%)
1	75		(18.6%)
2	113		(28%)
3	87		(21.5%)
4	51		(12.6%)
5	11		(2.7%)
6	8		(2%)
7	0		(0%)
8	0		(0%)
9	0		(0%)
EHRA I	48		(11.9%)
EHRA II	282		(69.8%)
EHRA III	67		(16.6%)
EHRA IV	7		(1.7%)
Primary intervention	302		(79.2%)
Re-do intervention	104		(20.8%)
Anticoagulation			
Vitamin K antagonists	23		(5.7%)
Thrombin inhibitors	16		(4%)
Factor Xa antagonists	329		(81.4%)
None	36		(8.9%)

**Table 2 jcm-11-02546-t002:** Complications.

Venous-Access-Related		
Hematoma *	10	2.4%
- AV-fistula	4	1%
- Pseudoaneurysm	5	1.2%
Thrombosis	1	0.2%
Infection	0	0%
Transseptal-access-related		
Aortic root puncture	0	0%
Aortic dissection	0	0%
Pericardial effusion	0	0%
- Tamponade	0	0%
Cardiac perforation	0	0%
Myocardial infarction	0	0%
Embolism	0	0%
- Stroke or transient ischemic attack	0	0%
- Peripheral arterial occlusion	0	0%
Ablation-related		
Pulmonary vein stenosis	0	0%
Phrenic nerve palsy	0	0%
Atrio-esophageal fistula	0	0%
Valvular lesions	0	0%
AV-Block	0	0%
TEE-related		
Gastroesophageal bleeding	1	0.2%
Gastroesophageal rupture	0	0%
Major complications		
Death	0	0%
Myocardial infarction	0	0%
Stroke	0	0%
Emergency vascular, cardiac or visceral surgery	0	0%

* Hematoma: defined as prolonged pressure bandage > 6 h, AV-fistula, pseudoaneurysm, retroperitoneal hematoma or inguinal hematoma requiring transfusion.

## Data Availability

In accordance with the restrictions of the ethics committee decision, all data are available for research purposes as encrypted anonymous files on request from the corresponding author.

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
