# Peer review of "Transesophageal Echocardiography-Guided Transseptal Left Atrial Access to Improve Safety in Patients Undergoing Pulmonary Vein Isolation"

_jcm, 2022, doi:10.3390/jcm11092546_

Round 1

Reviewer 1 Report

This is a retrospective study aimed to investigate the safety of TEE-guided trans-septal puncture during PVI in a single institution. The sample size was not prospectively defined to determine the likelihood of detecting rare complications (i.e. aortic puncture was reported to occure in 0.05 - 0.07% of cases https://doi.org/10.1093/europace/euw037), and no control group is available (the comparison was intended to be with the literature reports). 

The following could be helpful to make the paper more informative:

  1. The study consent was presumably retrospective? Please include the statement.
  2. The study is to establish safety of TEE-guided approach. Please state which contraindications were in place for performing TEE?
  3. The methods section contains detailed descripion of the ablation, but lacks any details on TEE. Please describe TEE protocol in the methods section, in particular, state whether full standard (ASE?) assessment was done prior to the ablation or only focused assessment of the IAS/LA/PV? Was only 2D or also 3D imaging used? Was 3D or Bi-plane imaging used to optimize septal puncture procedure? Please decsribe TEE insertion (blind or laryngoscopy-assisted). Describe TEE windows/views used at each stage of the procedure? I was unable to see Figure 1, panels A,C,D in the file provided, thus unable to comment on the Figure.
  4. Do the patients ineligible for TEE undergo PVI under general anaesthesia or conscious sedation? hat type of GA was used for patients included in your study (please describe in methods, including the use of muscle relaxants, topical anaesthesia for TEE insertion, the type of airways used, mouth guard etc). All of this may have an mpact on the amount of complications associated with TEE.
  5. The only complications of TEE reported in Table 2 were gastroesophageal rupture and bleeding. However, there are many other complications of TEE, including (but not limited to) sore throat, teeth injury, cardiac arrhythmias, inadvertent tracheal intubation, dysplacement of the ETT etc.). Did the authors attempt to identify these problems?
  6. Were there any anaesthetic complucations recorded?
  7. The second paragraph in the Results Section provides limited description of TEE-guided IAS puncture. It should be moved to the Methods section.
  8.  How was the esophageal haematoma diagnosed?
  9. Reduction in procedural time and radiation dose has been reported in Figures 2/3. I would suggest to report mean +/- SD in the results section and to add linear trend to the bars and whiskers plots in Figures 2-3.
  10. The word "sheath" is misspelled in the last sentence of Fig 1 legend.
  11. I would suggest to make stronger emphasis on TEE-guided vs non-TEE IAS puncture in the Discussion section. Perhaps, more literature-extracted data is needed to better demonstrate the advantage of TEE-guided procedure vs without TEE

Author Response

Response to Reviewer 1

The authors thank Reviewer 1 for his fast, careful and thorough feedback on our submission. We greatly acknowledge the comments which will help us to improve our paper.

Point-by-point reply:

  1. Consent was prospective on all procedures (in Switzerland, all data must be submitted to a central data base (CH -paceweb) for quality control). The study was additionally approved by the ethics committee (KEK ID 2018-00451). This is now clearly stated in the revised method section.
  2. Contraindications are now mentioned in the revised method section.
  3. The method section was entirely revised according to the reviewers suggestions. TEE procedure is now described in detail. In addition we apologize for the poor quality of figure 1 - panels were rearranged. In addition we submitted high resolution images of the four panels together with the revised manuscript.
  4. Anesthesia procedure is now described in the method section, as suggested
  5. Reviewer 1 raises an important point. In our study design we prespecified only bleeding, esophageal rupture, teeth injury and failure to insert the TEE probe as TEE specific major events. There was however not a single insertion failure, and also no teeth injury (this is now stated in the revised manuscript). Minor complaints such as lacerations or sore throat may be due to TEE and anesthesia and are difficult to grade appropriately. They had therefore not been assessed. Serious complications such as arrhythmias, endotracheal tube displacement, etc can be attributed to anesthesia as well as to TEE insertion/manipulation. We state now in the revised manuscript that there were no serious anesthesia related complications.
  6. See comment 5
  7. Done, as suggested
  8. CT-scan - this is now stated in the manuscript
  9. Mean +/- SD data on all procedures are now mentioned in the text
  10. The misspelling is now corrected
  11. As suggested by reviewer 1 we put more, but still diplomatic emphasis on the potential advantage of TEE guided puncture in the discussion section. All relevant studies regarding complications of PVI procedures are now cited in the manuscript.

Reviewer 2 Report

The authors should shorten the manuscript and specifically shorten the description of AF ablation and focus on the TEE guided transeptal.  The section on the ablation procedure can be greatly shortened. 

The images on Figure 1 are of poor quality. 

Figure 3 has no units on the y axis. 

Author Response

The authors thank Reviewer 2 for his/her encouraging feedback on our work. We greatly appreciate the fast review.

In the revised manuscript the description of the ablation procedure is now greatly shortened.

Together with the revised manuscript we submit high resolution panels A - D of figure 1.

The y axis in figure 3 reads now cGy x cm2

Round 2

Reviewer 1 Report

The paper feels much better, thank you. The flow and information presented makes now quite clear sense and does nor raise additional major questions.

The only question that perhaps should be addressed in the discussion (and possibly, numerically reported over the bars/whiskers in Figures 2/3 is the dynamic change in fluoroscopy time and dose. The results section stated, that fluoro time was 5.825 +/- 5.07 min (the shortest time then should be 0.755 minute = 45 sec?), and the same with fluoro dose 642.4+/-612.8 (lowest dose was 29.6 cGyxcm2 ?). These numbers suggests that in some patients fluoroscopy was not needed at all, and that TEE was sufficient to guide the entire procedure?

Author Response

We thank Reviewer 1 for his/her comments.

Reviewer 1 is absolutely right with his/her interpretation of our data: in parallel with the operators learning curve, fluoroscopy times drastically decreased. In certain cases there was even zero to minimal radiation exposure. In fact, the combined approach with TEE guidance and use of the electromagnetic mapping system meanwhile allows catheter placement and transseptal access without fluoroscopy in some cases. 

Nevertheless, we cannot omit minimal radiation exposure in most patients, because our standard operating protocol requires confirmation of the position of the ablation catheter tip in relation to the esophagus (with the help of either the TEE probe or a radiolucent gastric tube) before ablation at the posterior walls of the pulmonary veins.

As suggested by Reviewer 1, the point of the dynamic change in fluoroscopy doses and times is now discussed in the revised discussion section.